# Glucolipotoxic Stress-Induced Mig6 Desensitizes EGFR Signaling and Promotes Pancreatic Beta Cell Death

**DOI:** 10.3390/metabo13050627

**Published:** 2023-05-04

**Authors:** Yi-Chun Chen, Andrew J. Lutkewitte, Halesha D. Basavarajappa, Patrick T. Fueger

**Affiliations:** 1Department of Pediatrics and Herman B. Wells Center for Pediatric Research, Indiana University School of Medicine, Indianapolis, IN 46202, USA; 2Department of Cellular & Integrative Physiology, Indiana University School of Medicine, Indianapolis, IN 46202, USA; 3Department of Molecular and Cellular Endocrinology, Beckman Research Institute of the City of Hope, Duarte, CA 91010, USA

**Keywords:** islet, Errfi1, diabetes

## Abstract

A loss of functional beta cell mass is a final etiological event in the development of frank type 2 diabetes (T2D). To preserve or expand beta cells and therefore treat/prevent T2D, growth factors have been considered therapeutically but have largely failed to achieve robust clinical success. The molecular mechanisms preventing the activation of mitogenic signaling pathways from maintaining functional beta cell mass during the development of T2D remain unknown. We speculated that endogenous negative effectors of mitogenic signaling cascades impede beta cell survival/expansion. Thus, we tested the hypothesis that a stress-inducible epidermal growth factor receptor (EGFR) inhibitor, mitogen-inducible gene 6 (Mig6), regulates beta cell fate in a T2D milieu. To this end, we determined that: (1) glucolipotoxicity (GLT) induces Mig6, thereby blunting EGFR signaling cascades, and (2) Mig6 mediates molecular events regulating beta cell survival/death. We discovered that GLT impairs EGFR activation, and Mig6 is elevated in human islets from T2D donors as well as GLT-treated rodent islets and 832/13 INS-1 beta cells. Mig6 is essential for GLT-induced EGFR desensitization, as Mig6 suppression rescued the GLT-impaired EGFR and ERK1/2 activation. Further, Mig6 mediated EGFR but not insulin-like growth factor-1 receptor nor hepatocyte growth factor receptor activity in beta cells. Finally, we identified that elevated Mig6 augmented beta cell apoptosis, as Mig6 suppression reduced apoptosis during GLT. In conclusion, we established that T2D and GLT induce Mig6 in beta cells; the elevated Mig6 desensitizes EGFR signaling and induces beta cell death, suggesting Mig6 could be a novel therapeutic target for T2D.

## 1. Introduction

In normal physiological conditions, to ensure proper, robust insulin release, beta cell insulin secretory capacity (i.e., beta cell function) and insulin stores (dependent on beta cell mass) are tightly regulated to match demand and dynamically respond to a host of stressors. For example, during pregnancy and obesity, functional beta cell mass (the composite of beta cell mass and functional capacity) is increased by responding to the pressure to secrete more insulin to maintain optimal glycemic control. This dynamic ability to expand functional beta cell mass is in response to metabolic, hormonal, and cellular signaling cues [1]. Studies have suggested that growth factor receptor signaling pathways, such as epidermal growth factor receptor (EGFR) signaling, are among the contributors to beta cell mass expansion by promoting beta cell replication and survival [2,3]. Unfortunately, the capacity to increase and even preserve functional beta cell mass is finite and can be overwhelmed by sustained, exhaustive pressure [4].

The inability of beta cells to meet a sustained demand to produce and secrete insulin imposed by systemic insulin resistance precipitates type 2 diabetes (T2D), where de-compensation of functional beta cell mass occurs. In the period of pre-diabetes leading up to frank T2D, bouts of hyperglycemia and persistent hyperlipidemia (glucotoxicity and lipotoxicity, respectively, and hereafter jointly referred to as glucolipotoxicity, or GLT) accentuate beta cell failure. Interestingly, the activation of proximal signaling molecules of growth factor receptors (e.g., Akt and glycogen synthase kinase 3/GSK3) is inhibited in beta cells under stressed conditions [5,6,7]. Such observations suggest that activation of the growth factor receptors themselves might be inhibited during the development of T2D, thereby limiting beta cell proliferation and survival. Thus, we hypothesized that the glucolipotoxic milieu central to the development of T2D suppresses the activation of beta cell growth factor receptors, thereby inactivating survival signaling cascades and contributing to beta cell loss through apoptosis. Further, we speculate that endogenous feedback inhibition is a mechanism for the suppression of growth factor signaling in beta cells in T2D.

In many cells, EGFR signaling is engaged to promote cellular replication and survival; indeed, EGFR signaling is central to preserving beta cells in vivo and in vitro. Mice expressing constitutively-active EGFR in the pancreas are protected against the development of beta cell toxin-induced diabetes in that their beta cell loss is inhibited [8]. Conversely, pharmacologically blocking Raf-1 signaling downstream of EGFR induces beta cell death [9]. As alluded to above, EGFR signaling is also required for beta cell expansion, as mice expressing a dominant negative form of EGFR fail to acquire compensatory beta cell expansion during pregnancy and obesity [2,3]. Thus, the inactivation of EGFR signaling has dire consequences for functional beta cell mass. Nevertheless, little attention has been given to the molecular mechanisms controlling the inactivation of growth factor signaling in the context of T2D.

In most signaling cascades, the activation of a pathway is followed by inactivation through classical negative feedback; EGFR signaling is no different. We have investigated the impact of endogenous feedback inhibition of EGFR by the adapter protein Mig6 on functional beta cell mass [10,11]. Here, we report that GLT induces Mig6 where it terminates EGFR signaling and promotes beta cell apoptosis. Thus, suppressing the actions of Mig6 could be fruitful for re-engaging pro-survival signaling in beta cells in T2D.

## 2. Materials and Methods

### 2.1. Cell Culture, Reagents, and the Use of Adenovirus

INS-1-derived 832/13 and 828/33 rat insulinoma cells (kindly provided by Dr. Christopher Newgard, Duke University) were grown in 11.1 mM D-glucose RPMI-1640 medium supplemented with 10% fetal bovine serum, 100 U/mL penicillin, 100 μg/mL streptomycin, 10 mmol/L HEPES, 2 mmol/L L-glutamine, 1 mmol/L sodium pyruvate, and 50 μmol/L β-mercaptoethanol, as previously described [12,13]. 832/13 cells are glucose-responsive for secreting insulin and sensitive to apoptosis. 828/33 cells stably overexpress Bcl-2 and are thus resistant to apoptosis.

In the lipotoxicity experiments, sodium palmitate (Sigma; St. Louis, MO, USA) was dissolved in 0.1M NaOH buffer at 70 °C and mixed with 5% fatty acid-free BSA (Sigma) solution at 37 °C to yield a 5 mM palmitic acid-BSA complex stock solution. The palmitic acid-BSA complex was diluted into a serum-free RPMI culture medium to obtain various concentrations of palmitic acid ranging from 0.1 to 0.4 mM. In glucotoxicity experiments, 25 mM glucose and serum-free RPMI culture medium supplemented with 0.1% BSA was used as a high-glucose treatment, and 5 mM glucose plus 20 mM D-mannitol (osmotic control; Sigma) was used as the low-glucose control. For glucolipotoxicity experiments, 0.4 mM palmitic acid plus 25 mM glucose medium was used to create glucolipotoxicity.

For EGF stimulation experiments, 832/13 cells were challenged with glucolipotoxicity for 4 h, starved in RPMI 1640 medium containing 2.5 mM glucose and 0.1% BSA for 2 h, and treated with 10 ng/mL rat recombinant EGF (R&D Systems) for 5 min. For growth factor stimulation experiments, 832/13 cells were starved for 2 h and treated with insulin-like growth factor 1 (IGF-1, NIH repository) or hepatocyte growth factor (HGF, R&D Systems) for various times.

In our gene overexpression studies, recombinant adenoviral vectors expressing Mig6 or green fluorescent protein (GFP) under the control of cytomegalovirus (CMV) promoter were used as previously described [11]. In our gene suppression studies, recombinant adenoviral vectors expressing small interfering RNAs (siRNA) specific to rat Mig6 or with no known gene homology (scrambled siRNA) were used as previously described [14]. Alternatively, for cell apoptosis experiments, Mig6 siRNA (Mig6 ON-TARGET plus Smartpool, Dharmacon) was transfected with Lipofectamine 2000 (Invitrogen) to suppress Mig6 expression, and non-targeting siRNA served as a negative control. Cells were challenged by glucolipotoxicity 72 h after transfection, and apoptosis assays were performed.

### 2.2. Human and Rodent Islet Experiments

Cadaveric human islets were obtained from the Integrated Islet Distribution Program or the Southern California Islet Resource Center affiliated with the Beckman Research Institute of the City of Hope. Islets from four donors with T2D and four donors without diabetes were analyzed to determine *MIG6* expression levels in mRNA isolated from islets upon receipt to the laboratory (typically one day after isolation). In signaling experiments, human islets from eight donors with BMIs lower than 30 were cultured similarly to cell experiments with the exception that CMRL-1066 media with 5 mM glucose and supplemented with 10% fetal bovine serum, 50 units/mL penicillin, and 50 µg/mL streptomycin was used.

Rat pancreatic islets were collected from male Wistar rats weighing approximately 250 g [15,16]. After collagenase digestion, islets were hand-picked and cultured in 5 mM glucose RPMI medium supplemented with 10% fetal bovine serum, 50 units/mL penicillin, and 50 µg/mL streptomycin overnight before drug treatments.

### 2.3. Apoptosis Assays

Following siRNA treatment (48 h), 40,000 832/13 cells were plated in black-walled, 96-well plates. The following day, media was replaced with starvation media, and cells were treated with either BSA or palmitic acid-coupled BSA for the times indicated. Caspase 3/7 activity was measured using an Apolive-Glo kit (Promega, Madison, WI, USA) and measured using a SpectraMax M5 (Molecular Devices, Sunnyvale, CA, USA). Briefly, following GLT treatments, cells were incubated with Caspase-Glo 3/7 Reagent for 30 min at RT, and luminescence was measured.

### 2.4. Immunoblot Analysis

Cells were lysed in 1% IGEPAL reagent supplemented with 10% glycerol, 16 mM NaCl, 25 mM HEPES, 60 mM n-octylglucoside (Research Products International Corp., Mt. Prospect, Illinois, USA), phosphatase inhibitor cocktails (PhosSTOP tablets, Roche), and protease inhibitor cocktails (EDTA-free cOmplete tablets, Roche). Lysates were resolved on a 10% NuPAGE Bis-Tris Gel (Invitrogen, Waltham, MA, USA), transferred to an Immobilon-FL Transfer Membrane (Millipore, Burlington, MA, USA), and incubated with primary antibodies (Table 1). Subsequently, membranes were incubated with IRDye 800 or 700 fluorophore-labeled secondary antibodies from LI-COR. Protein bands were visualized using the Odyssey System (LI-COR) and quantified with Image J software (NIH). Phosphorylated protein levels were normalized to the total protein levels in the cell lysate, and the total (e.g., non-phosphorylated) protein levels were normalized to tubulin or GAPDH protein levels.

### 2.5. Quantitative RT-PCR Analysis

RNA from 832/13 cells, rat islets, and human islets were isolated using RNeasy Mini or Micro kits (Qiagen; Valencia, CA, USA). Reverse transcription was completed with a High Capacity cDNA Reverse Transcription kit (Applied Biosystems, Waltham, MA, USA). The threshold cycle methodology was used to calculate the relative quantities of the mRNA products of *Mig6, Socs4, Socs5, Frs3, and Lrig1* (Table 2). PCR reactions were performed in triplicate for each sample from at least three independent experiments and were normalized to *Gapdh* or *beta-actin* gene expression levels.

### 2.6. Statistical Analysis

All data are reported as means ± SEM. Protein and mRNA data were normalized to control conditions and were presented as relative expressions. Student’s *t*-test or ANOVA (with Bonferroni post hoc tests) were performed using GraphPad Prism software to detect statistical differences (*p* < 0.05).

## 3. Results

### 3.1. GLT and ER Stress Attenuate EGFR Activation in Rodent Beta Cells and Human Islets

To study the extent to which GLT compromises EGFR activation, we exposed human islets and 832/13 INS-1 cells to a medium containing high glucose and palmitic acid and assessed the phosphorylation of EGFR. We identified that GLT treatment prevents EGF-mediated EGFR phosphorylation in both human islets and the beta cell line (Figure 1). Notably, neither the basal phosphorylation nor total cellular abundance of EGFR was changed by GLT (data not shown), indicating that the attenuated EGFR phosphorylation and activation were likely the consequences of EGFR kinase interruption. Because GLT imposes endoplasmic reticulum (ER) stress on beta cells and triggers deleterious effects such as beta cell death, indicated by phosphorylation of eIF2α and JNK as well as elevated cleaved caspase 3 (Figure 1A and Figure 2A,B), we examined the extent to which ER stress induction alone was sufficient to attenuate EGFR activation. Pretreatment with thapsigargin (the sarcoendoplasmic reticulum Ca(2+) ATPase 2 pump inhibitor) induces ER stress and inhibits EGFR activation in 832/13 cells (Figure 2C,D). The above findings suggest that the pathological stress stimuli present in T2D compromise the activation of EGFR. Importantly, cell death in GLT per se does not decrease EGFR phosphorylation, as beta cells overexpressing Bcl-2 (828/33 INS-1 cells), which confers resistance to apoptosis, were still sensitive to GLT with respect to its ability to prevent maximal EGFR phosphorylation (Figure 3).

### 3.2. EGFR Feedback Inhibitor Mig6 Is Elevated in GLT-Treated Beta Cells and T2D Human Islets

To identify the factors associated with EGFR inactivation during GLT, we examined a set of well-defined, inducible EGFR inhibitors [17]. First, we discovered that Mig6, an adaptor protein that blocks EGFR activation, was induced by GLT, whereas expression of other EGFR inhibitors—SOCS4, SOCS5, FRS3, and LRIG1—all remained unchanged by GLT (Figure 4). To further study the specific effects of glucotoxicity and lipotoxicity on the induction of Mig6, we treated the 832/13 cells with high levels of glucose and/or palmitic acid and measured Mig6 expression levels (Figure 5). We established that high glucose induced Mig6 in a dose-dependent manner that was not due to the osmotic actions of glucose (Figure 5A,B). However, palmitic acid alone did not induce Mig6 at the concentrations and time points examined (Figure 5C,D). In addition, because high glucose stimulates insulin secretion in the beta cells, we needed to determine the extent to which the autocrine or paracrine effects of insulin might promote Mig6 expression; exogenous insulin treatment did not alter Mig6 expression levels (Figure 5E). Finally, we identified that Mig6 mRNA expression was greater in islets isolated from donors with T2D compared to control donors, and Mig6 mRNA and protein expression was induced in rodent islets exposed to GLT (Figure 6). These data suggested that diabetogenic stress induces Mig6 in beta cells.

### 3.3. Mig6 Inhibits EGFR in Pancreatic Beta Cells and Promotes Death during GLT

Because GLT hinders EGFR activation and the inhibitor of EGFR, Mig6, is induced by GLT, it is intuitive to speculate that stress-inducible Mig6 controls EGFR inactivation during GLT. Thus, we used an RNA interference approach to examine the functional significance of Mig6 in EGFR signaling. Suppression of Mig6 enhanced both EGFR and ERK1/2 phosphorylation following EGF stimulation (Figure 7A–C). In contrast, elevated Mig6 expression dampened downstream ERK1/2 phosphorylation (Figure 7D–F). Importantly, the actions of Mig6 were restricted to EGFR signaling, as altering Mig6 expression (overexpression or suppression) did not alter hepatocyte growth factor-stimulated ERK1/2 phosphorylation or insulin-like growth factor-1-stimulated Akt phosphorylation (Figure 7G,H).

Finally, as stress-induced Mig6 suppresses the EGFR signaling pathway, we sought to determine the extent to which silencing Mig6 would restore EGFR activity and prevent beta cell death during GLT. Again, using siRNA-mediated suppression of Mig6 (Figure 8A), we determined that reducing Mig6 increased EGFR and ERK1/2 phosphorylation during GLT (Figure 8B–D). Importantly, Mig6 suppression limited beta cell apoptosis in GLT, as measured by caspase 3/7 activity (Figure 8E).

## 4. Discussion

Although genetic manipulation of pancreatic EGFR in mice leads to the acceleration or prevention of diabetes, the natural history of EGFR kinase activity during different phases of the progression to diabetes remains unknown [2,3,8,18]. In regard to beta cell de-compensation, the phase of declining functional beta cell mass prior to the onset of frank T2D, the extent to which diabetogenic stress stimuli alter EGFR activity and impact beta cell life/death decisions is unclear. It has been reported that diabetic stressors could compromise the activation and propagation of receptor tyrosine kinase (RTK) signaling cascades in pancreatic beta cells [19,20]. For example, GLT and cytokine challenges hinder the activation of insulin receptors and downstream phosphatidylinositol 3-kinase, hence preventing the cytoprotective effects of insulin in beta cells [5,7,21]. However, the molecular mechanisms responsible for this stress-mediated RTK inactivation remain to be defined. It is likely that there are stress-responsive factors that crosstalk with RTK signaling machinery in pathological conditions.

In this study, we demonstrated that glucolipotoxicity and ER stress attenuate EGFR activation in pancreatic beta cells via the stress-responsive EGFR inhibitor, Mig6. Mig6 was initially characterized as an endogenous EGFR feedback inhibitor but has also been suggested to impair other RTKs [22,23,24], yet in our work here, we have been unable to ascribe the actions of Mig6 to HGF or IGF-1 signaling in 832/13 rat pancreatic beta cells. After mitogen stimulation, Mig6 is activated to abolish EGFR signaling transmission via a two-tiered mechanism: (1) Mig6 binds to the EGFR intracellular kinase domain and inhibits kinase dimerization and activation, and (2) Mig6 facilitates EGFR endo-lysosomal sorting and degradation [21]. However, a new role of Mig6 as a stress-induced modulator of cellular signaling and function has also been revealed. Makkinje et al. first reported that mechanical stress in diabetic nephropathy is sufficient to induce Mig6, and the transient expression of Mig6 results in selective activation of JNK [25]. Later, Mabuchi et al. further suggested that Mig6 is able to bind to I kappa B alpha, resulting in NF-kappa B activation [26]. Additionally, Hopkins et al. demonstrated that ligand deprivation promotes Mig6-mediated c-Abl activation and cell death [27]. Other work has suggested that Mig6 modulates the DNA damage response by interacting with the serine/threonine kinase ATM and histone H2AX [28]. Furthermore, as previously reported by our group, both ER stress and pro-inflammatory cytokines induce Mig6, and haploinsufficiency of Mig6 prevents mice from developing an experimentally induced form of T1D [10,29].

Here, we established that GLT-induced beta-cell apoptosis is mediated, at least in part, by the induction and actions of Mig6. The deleterious Mig6-mediated effects could be EGFR-dependent and/or independent. Beyond the well-described feedback inhibition of EGFR signaling, Mig6 activates pro-apoptotic JNK via its Cdc42/Rac interactive binding domain, representing an EGFR-independent response [25]. In addition to Mig6, there are likely other stress-mediated factors controlling EGFR inactivation and downstream alterations in ERK1/2 signaling. For instance, cell surface EGFR could be modified and inhibited by advanced-glycation precursors present in GLT, and cellular stress-activated phosphatases could also inactivate EGFR [30,31,32]. Importantly, we established that several other feedback inhibitors of EGFR—Socs4, Socs5, Frs3, and Lrig1—were not induced by GLT in beta cells, thus highlighting the importance of Mig6 in EGFR feedback inhibition. ERK1/2 signaling can also be modulated by factors others than EGFR in the context of GLT, and thus pathways beyond EGFR-Mig6 must be considered. Nevertheless, we have provided evidence that the feedback inhibition of EGFR in GLT is likely mediated by the direct actions of Mig6.

This work presents a potentially novel mechanism for reduced beta cell proliferation and survival in the states of chronic over-nutrition that trigger beta cell stress. It is known that short-term intralipid infusion enhances beta cell proliferation via EGFR and mTOR signaling pathways in adult rodents [33], but chronic nutrient overload (i.e., high-fat diet feeding) does not promote beta cell mass expansion [34]. The contribution of Mig6 to restraining beta cell proliferation and survival during obesity in vivo remains to be determined. However, as Mig6 is elevated in islets isolated from T2D patients, we speculated that Mig6 perhaps contributes to the dampening of EGFR signaling activation during the progression of T2D.

Despite the advances in our understanding of the effects of GLT on the beta cell in the current study, there are some caveats and limitations worth noting. First, most studies examined the impact of combined elevated glucose and lipid (i.e., GLT) on beta-cell signaling and viability. Defining the specific effects of gluco- or lipotoxicity that are attributed to Mig6 warrants further study. Similarly, whereas the effects of beta cell stress on Mig6 expression were examined in both primary rodent and human islets, extending the signaling work to islets with the use of potential inhibitors of Mig6 is essential for further examining the utility of Mig6 as a target. Finally, most of the actions of Mig6 have been viewed through the lens of its role as a feedback inhibitor of EGFR. However, as an adapter protein, Mig6 could interact with other factors important for beta cell signaling and viability. Indeed, we recently discovered a novel interaction between Mig6 and NumbL, an adaptor protein and negative regulator of Notch signaling [35]. Thus, the actions of Mig6 are likely more complicated than strictly feedback inhibition of EGFR.

## 5. Conclusions

In summary, we discovered that GLT attenuates EGFR activation via Mig6, and Mig6 modulates GLT-induced beta cell apoptosis. This work highlights the broad effects that a diabetogenic milieu has on cellular signaling and suggests that reactivation of pro-survival RTK signaling could be beneficial for fortifying functional beta cell mass. How Mig6 might control beta cell survival beyond the direct feedback inhibition of EGFR remains to be determined. Nevertheless, this work highlights the potential of targeting adaptor proteins and feedback inhibitors as a means to prevent or reverse diabetes. Thus, we propose that Mig6 might be a suitable therapeutic target to promote beta cell survival in preventing and treating T2D.

## Figures and Tables

**Figure 1 metabolites-13-00627-f001:**
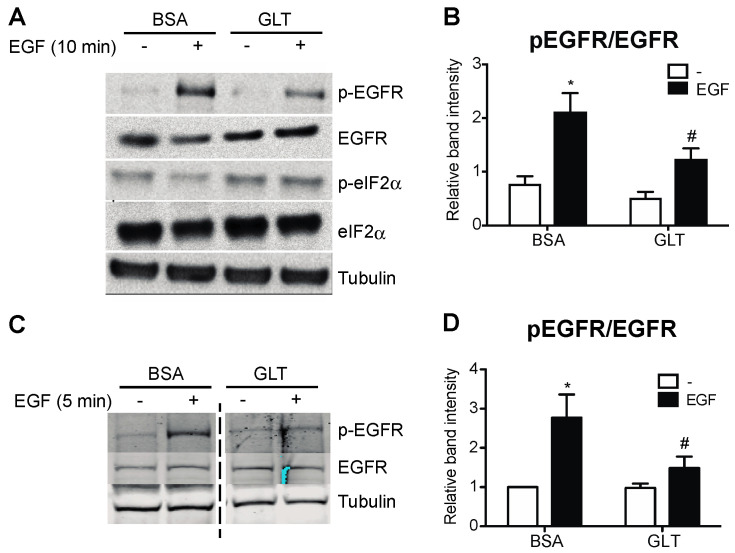
Glucolipotoxicity impairs EGFR activation in beta cells and human islets. Human islets and 832/13 cells were cultured in media with 5 mM glucose and BSA, or 25 mM glucose and 400 μM palmitic acid complexed to BSA (glucolipotoxicity, GLT) for 48 or 8 h, respectively, followed by starvation in 5 mM glucose medium for 2 h, and then stimulated with recombinant human EGF (50 ng/mL for 10 min for islets and 10 ng/mL for 5 min for cells). Protein levels of p-EGFR, EGFR, p-eIF2α, and tubulin were analyzed by immunoblotting. Shown are representative immunoblots from human islets (**A**,**B**) and 832/13 cells (**C**,**D**), and quantified data are reported as fold induction relative to BSA, non-stimulated samples. Groups were compared using ANOVA with Bonferroni post hoc tests. n = 3 independent human islet preparations and 3 independent cell line experiments; * *p* < 0.05 vs. BSA, non-stimulated; # *p* < 0.05 vs. BSA, EGF-stimulated.

**Figure 2 metabolites-13-00627-f002:**
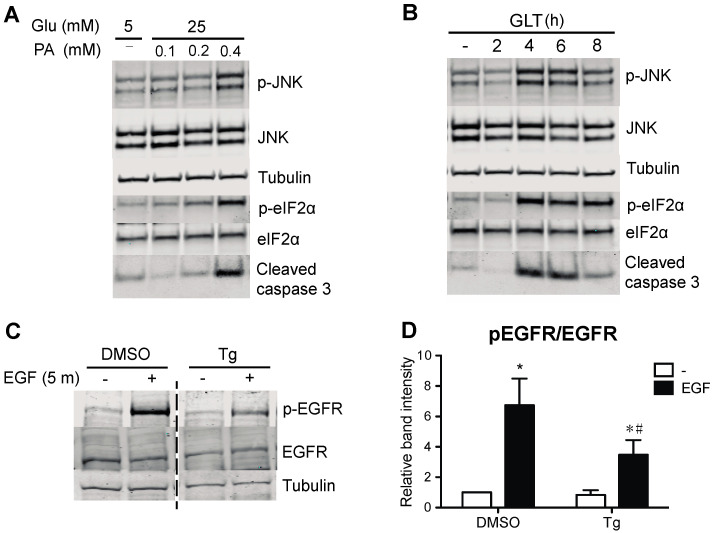
ER stress impairs EGFR phosphorylation. (**A**) To verify induction of ER stress, 832/13 cells were cultured in media containing 5 mM glucose and BSA or 25 mM glucose and increasing concentrations of palmitic acid complexed to BSA (0, 0.1, 0.2, or 0.4 mM) for 8 h. (**B**) In a complementary experiment, cells were exposed to 25 mM glucose and 0.4 mM palmitic acid for up to 8 h. Cells were harvested, and lysates were immunoblotted using antibodies directed against p-JNK, JNK, p-eIF2α, eIF2α, and cleaved caspase 3 to establish the extent of ER stress produced by glucolipotoxicity. Shown are representative, confirmatory immunoblots. (**C**) To induce ER stress, 832/13 cells were treated with DMSO (vehicle control) or 1 μM thapsigargin (Tg) for 4 h, followed by starvation and EGF stimulation as before. Protein levels of p-EGFR, EGFR, and tubulin were analyzed by immunoblotting. Representative blots of n ≥ 3 experiments are shown, and results are quantified in (**D**). Groups were compared using ANOVA with Bonferroni post hoc tests. * *p* < 0.05 vs. BSA, non-stimulated; # *p* < 0.05 vs. BSA, EGF-stimulated.).

**Figure 3 metabolites-13-00627-f003:**
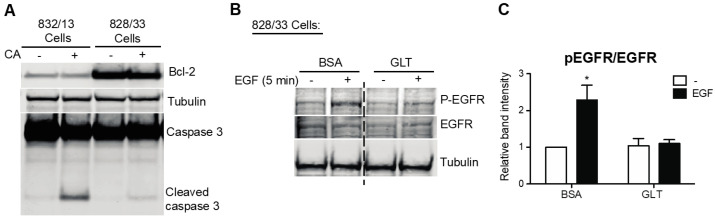
Apoptosis is not required for glucolipotoxicity-impaired EGFR activation. (**A**) To promote apoptosis, 832/13 and Bcl-2 overexpressing 828/33 cells were treated with 1 μM camptothecin (CA) for 8 h. To demonstrate resistance to cell death in 828/33 cells, protein levels of Bcl-2, full-length and cleaved caspase 3, and tubulin were analyzed by immunoblotting. (**B**) A total of 828/33 cells were cultured in media, as in Figure 1. Protein levels of p-EGFR, EGFR, and tubulin were analyzed by immunoblotting, and quantified results are reported in (**C**). Groups were compared using ANOVA with Bonferroni post hoc tests. n = 3; * *p* < 0.05 vs. all other groups.

**Figure 4 metabolites-13-00627-f004:**
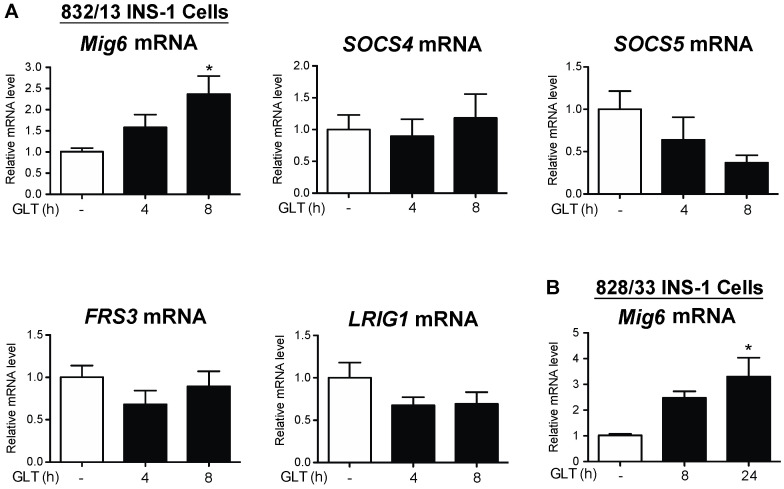
Mig6, but not other feedback inhibitors of EGFR, is induced by glucolipotoxicity. (**A**) 832/13 and (**B**) 828/33 cells were cultured in control (white bars) or glucolipotoxic (black bars) media for up to 8 h. Expression of Mig6, SOCS4, SOCS5, FRS1, and LRIG1 mRNA was quantified using qRT-PCR. Groups were compared using ANOVA. n = 3; * *p* < 0.05 vs. control media.

**Figure 5 metabolites-13-00627-f005:**
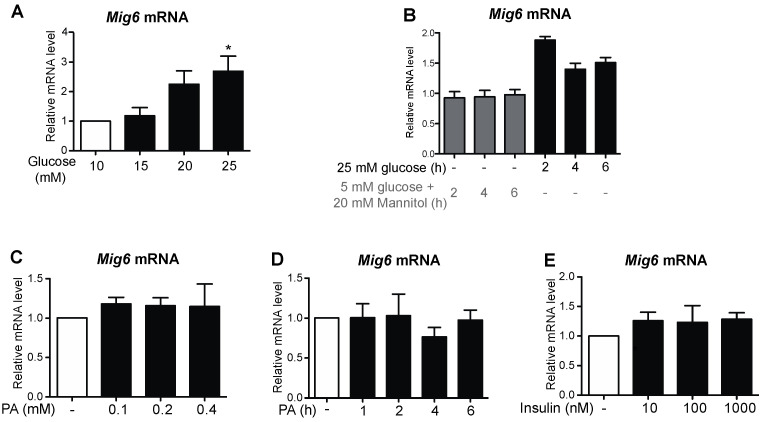
Gluco-, but not lipotoxicity, alone induces Mig6 expression. 832/13 cells were treated with (**A**) 5, 10, 15, 20, or 25 mM glucose for 4 h, (**B**) 25 mM glucose or 5 mM glucose + 20 mM mannitol (as an osmotic stress control) for 0, 2, 4, or 6 h. (**C**) BSA, 100, 200, 400 μM palmitic acid complexed to BSA for 4 h. (**D**) A total of 400 μM palmitic acid for the indicated times, or (**E**) 0, 10, 100, or 1000 nM recombinant human insulin for 4 h. Mig6 mRNA levels were determined by qRT-PCR. Groups were compared using ANOVA. n ≥ 3 experiments. * *p* < 0.05 vs. BSA/5 mM glucose + 20 mM mannitol.

**Figure 6 metabolites-13-00627-f006:**
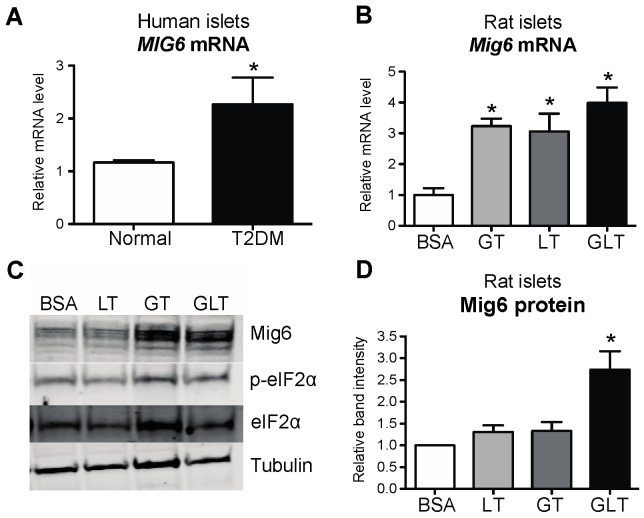
Mig6 is elevated in T2DM human and rodent islets treated with glucolipotoxicity. (**A**) MIG6 mRNA was measured in human islets from normal (white bar) and type 2 diabetic (black bar) cadaver donors. (**B**) Rat islets were cultured in media containing 5 mM glucose and BSA (white bar), 25 mM glucose and BSA (glucotoxicity, GT; light gray bar), 5 mM glucose and 0.4 mM palmitic acid complexed to BSA (lipotoxicity, LT; dark gray bar), or 25 mM glucose and 0.4 mM palmitic acid (glucolipotoxicity, GLT; black bar) for 8 h. Mig6 mRNA was measured by RT-PCR. (**C**) Rat islet lysates were immunoblotted with antibodies directed against Mig6, p-eIF2α, eIF2α, and tubulin, and (**D**) results for Mig6 content were quantified. Groups were compared using ANOVA with Bonferroni post hoc tests. N = 3–4; * *p* < 0.05 vs. normal or 5 mM glucose with BSA.

**Figure 7 metabolites-13-00627-f007:**
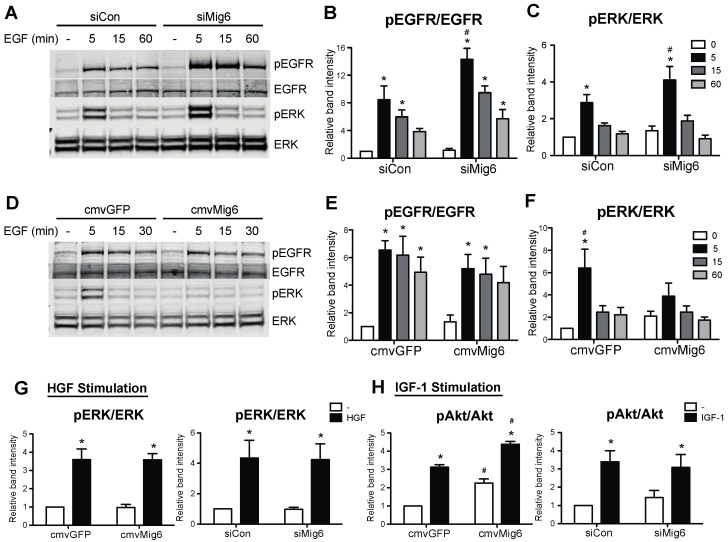
Mig6 controls EGF, but neither IGF1 nor HGF, pro-survival signaling pathways. (**A**–**F**) 832/13 cells were transduced with adenoviruses carrying cmvGFP vs. cmvMig6 or siCon vs. siMig6. Post transduction, cells were starved in 5 mM glucose and 0.1% BSA medium for 2 h, followed by 10 ng/mL recombinant rat EGF stimulation for 5 min. (**G**,**H**) After adenoviral transduction and starvation as in ((**A**,**D**), 832/13 cells were treated with recombinant human HGF or IGF-1 for 5 min. Protein levels of p-EGFR, p-Erk, Erk, p-Akt, Akt, and tubulin were analyzed by immunoblotting. Groups were compared using ANOVA with Bonferroni post hoc tests. n ≥ 3. * *p* < 0.05 vs. non-stimulated. # *p* < 0.05 vs. control virus, stimulated condition.

**Figure 8 metabolites-13-00627-f008:**
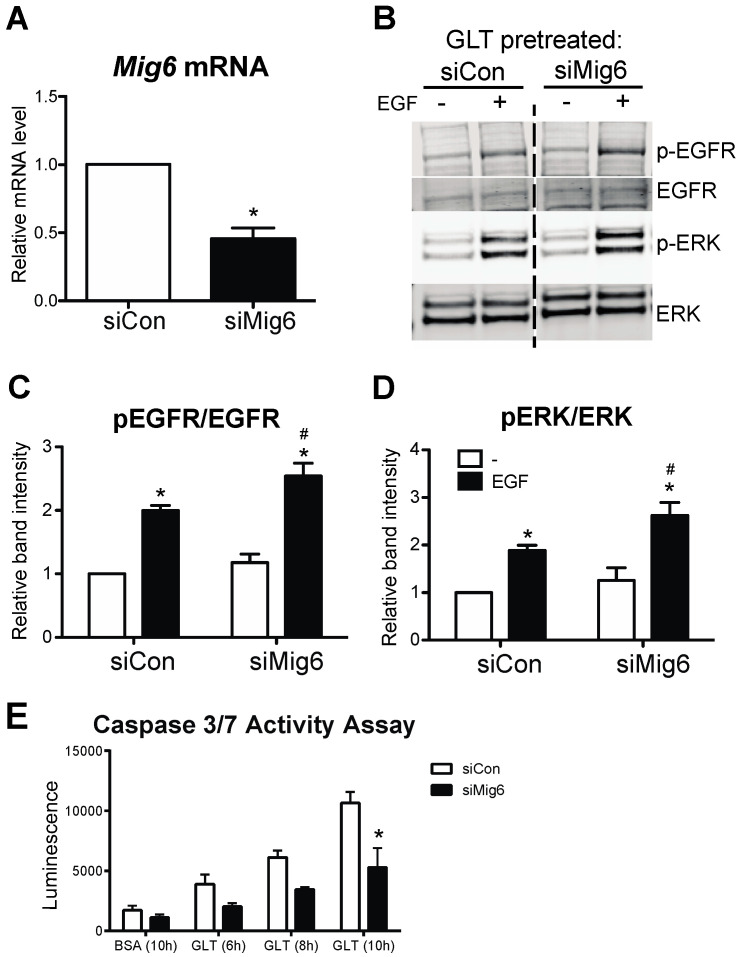
Mig6 suppression dampens apoptosis during glucolipotoxicity. (**A**) 832/13 cells were transduced with adenoviral vectors carrying either a scrambled control siRNA (siCon) or shRNA sequence against Mig6 (siMig6). Mig6 mRNA levels were determined by qRT-PCR. Groups were compared using Student’s *t*-test. n = 4; * *p* < 0.05. (**B**–**D**) Transduced cells were treated with GLT and EGF, as described previously. Protein levels of p-EGFR, EGFR, p-Erk, and Erk were determined by immunoblotting. Data are reported as fold induction related to the GLT-treated, non-EGF-stimulated group. Groups were compared using ANOVA with Bonferroni post hoc tests. n ≥ 3. * *p* < 0.05 vs. EGF-treated. # *p* < 0.05 vs. siCon EGF-stimulated. (**E**) Caspase 3/7 activity was measured following exposure to glucolipotoxic conditions in 832/13 cells siCon or siMig6. Groups were compared using ANOVA with Bonferroni post hoc tests. n = 3 experiments; * *p* < 0.05 vs. siCon.

**Table 1 metabolites-13-00627-t001:** List of antibodies used for immunoblotting.

Name	Vendor, Model Number	Dilution
Anti-Actin	MP Biomedicals, #691002	1:5000
Anti-Akt	Cell Signaling, #2920	1:1000
Anti-caspase 3	Cell Signaling, #9662	1:1000
Anti-CHOP	Santa Cruz, #7351	1:250
Anti-EGFR	Sigma-Aldrich, #E3138	1:1000
Anti-eIF2α	Cell Signaling, #5324	1:1000
Anti-ERK1/2	Cell Signaling, #4696	1:1000
Anti-γ-tubulin	Sigma-Aldrich, #T6557	1:5000
Anti-GAPDH	Abcam, #Ab9483	1:5000
Anti-Mig6	Santa Cruz, #D-1	1:250
Anti-phospho-Akt (Thr308)	Cell Signaling, #4056	1:1000
Anti-phospho-EGFR (Tyr1068)	Cell Signaling, #3777	1:1250
Anti-phospho-eIF2α (Ser51)	Cell Signaling, #3398	1:1000
Anti-phospho-ERK1/2 (Thr202/Tyr204)	Cell Signaling, #4370	1:2000
IRDye 800 or 700 fluorophore-conjugated antibodies	LI-COR	1:10,000

**Table 2 metabolites-13-00627-t002:** List of Taqman gene expression assays.

Name	Vendor	Assay ID
Rat Mig6	ThermoFisher Scientific	Rn01520744_g1
Human errfi1 (Mig6)	ThermoFisher Scientific	Hs00219060_m1
Rat Socs4	ThermoFisher Scientific	Rn01414734_m1
Rat Socs5	ThermoFisher Scientific	Rn01769079_m1
Rat Frs3	ThermoFisher Scientific	Rn01512038_m1
Rat Lrig1	ThermoFisher Scientific	Rn01421201_m1

## Data Availability

Data is contained within the article; raw data will be made available upon request to the corresponding author.

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
