# Peer review of "Glucolipotoxic Stress-Induced Mig6 Desensitizes EGFR Signaling and Promotes Pancreatic Beta Cell Death"

_metabolites, 2023, doi:10.3390/metabo13050627_

Round 1

Reviewer 1 Report

The paper explores the role of Mig6 as desensitizer of EGFR signaling pathway and the possibility that might be a therapeutic target for type 2 diabetes. The authors made a good contextualization of the subject and the research. The matters used were adequate and the results obtained are according to them. However, the statistical analysis is sparsely described. It is not clear if the author made the normalization tests for the results obtained. The authors should explain.

The paper has not an individual conclusion section, which I suggest to include in order to valorize it. The references must be written according to the journal rules.

Author Response

Thank you for the kind comments.  We have addressed your two concerns regarding statistical analysis and conclusion section.  First, we have added more details regarding the statistical analysis.  Second, we have added a separate Conclusions section in the revised manuscript.

Reviewer 2 Report

Data presented in figure 1: where is the 25mM glucose only control. it is not clear that BSA was presented in GLT conditions. authors need to clarify this. why was 25mM glucose only condition missing here?

same issue in Figure 2A, how can authors compare between 5mM glucose (normal glucose) to 25mM glucose (Diabetic) with varying doses of PA. include a 25mM glucose condition.

figure 2C, mention that DMSO is vehicle.

what was the test used in all comparisons for all figures.

is the similar lipotoxicity observed when glucose levels are at 5mM?

Author Response

Below are the comments from the reviewer and our response to them, which include additional information.

1. Data presented in figure 1: where is the 25mM glucose only control. it is not clear that BSA was presented in GLT conditions. authors need to clarify this. why was 25mM glucose only condition missing here?

  - We have clarified in the figure legends that palmitic acid is complexed to BSA for all of the experiments where lipid is used.

     This aspect of the response applies to comments 1, 2, and 5.  Whereas numerous studies have focused on the deleterious actions of glucotoxicity or lipotoxicity alone on the beta cell, we did not focus on either stressor alone in the present work for several reasons.  First, lipotoxicity alone did not increase Mig6 expression.  In beta cell lines at least, glucotoxicity, but not lipotoxicity, induced Mig6 expression.  Interestingly, in rat islets, the combination of glucolipotoxicity was required to increase Mig6 protein expression.  Thus, we focused most studies on glucolipotoxicity, as both of these event (i.e., hyperglycemia and hyperlipidemia) occur in type 2 diabetes.  Nevertheless, we have added this caveat or limitation to the discussion.  We hope the reviewer can accept why we focused on GLT to be able to keep focused on a milieu where Mig6 is induced.  We certainly agree that elevated glucose alone or lipid alone can be toxic to beta cells.

2. same issue in Figure 2A, how can authors compare between 5mM glucose (normal glucose) to 25mM glucose (Diabetic) with varying doses of PA. include a 25mM glucose condition.

  - Please see response to comment #1.

3. figure 2C, mention that DMSO is vehicle.

  - We have made this correction to the figure legend.

4. what was the test used in all comparisons for all figures.

  - We have added details regarding the statistical analysis in the figure legends.

5. is the similar lipotoxicity observed when glucose levels are at 5mM?

  - Please see response to comment #1.

Reviewer 3 Report

This manuscript is interesting, which describes in detail the role of stress-inducible EGFR inhibitor, Mig6 that regulates β-cell fate in a T2DM condition in which GLT induces Mig6, and thereby blunting EGFR signaling cascades, and mediates molecular events that regulate the β-cell. Overall, the manuscript is good and the authors have performed several experiments on in vitro and in vivo and prove their notion through the confirmation of their hypothesis via Apoptosis assays, Immunoblot analysis, and qRT-PCR analysis. The key findings also support their hypothesis and also in accordance with the already published reports. As a reviewer, this paper is overall good expect there are several grammatical mistakes and syntax that must be removed before its final consideration.

Author Response

Thank you for the kind comments on our manuscript.  We have carefully reviewed the manuscript and corrected grammatical errors and syntax as requested.

Round 2

Reviewer 2 Report

Thank you for making edits and providing explanations.